# METATST: ESSENTIAL TRANSFORMER COMPONENTS FOR TIME SERIES ANALYSIS

## ABSTRACT

This paper presents MetaTST, a versatile time series Transformer architecture that combines standard Transformer components with time series-specific features, omitting the traditional token mixer in favor of non-parametric pooling operators. The study's two primary contributions include defining the MetaTST architecture and showcasing its empirical success across forecasting, classification, imputation, and anomaly detection tasks. These results establish MetaTST as a robust and adaptable foundation for future time series Transformer designs, raising important questions about the necessity of attention mechanisms in time series analysis.

## 1 INTRODUCTION

Time series analysis techniques is widely used in real world applications. In recent years, deep learning for time series analysis has received great interests. Many classical models, such as MLP, CNN and RNN, have found their variations for time series analysis. Transformer (Vaswani et al., 2017), which is designed for NLP tasks, is now becoming popular in many areas such as CV (Dosovitskiy et al., 2021) and time series analysis. Benifits from its self-attention mechanism, Transformers can capture dependecies of long sequence. This lead to the success of Transformers in many areas.

In those time series transformers, Autoformer (Wu et al., 2021), FEDformer (Zhou et al., 2022) are among the best variants successfully applied to time series data. One of the main challenges they all trying to solve is the computation/memory bottleneck brought by the quadratic complexity of attention mechanism. With the insight that attention on time series often turns out to be sparse (Zhou et al., 2021), they adopt various substitute attention block specially designed for time series which can capture new time series features and have lower complexity. For example, the auto-correlation (Wu et al., 2021) replaces self-attention with series-wise connections that can be calculated efficiently via FFT (Fast Fourier Transform) with $O(L \log L)$ complexity. FEDformer use FFT and Wavelet Transform to capture the features in frequency demain. Along this line of research, the success of these models are mainly attributed to their newly devised attention substitution.

Although the performance of time series Transformers grows, its effectiveness is questioned by a recent work (Zeng et al., 2023). The authors demonstrate that a simple linear projection with seasonal-trend decomposition can outperform most Transformer variants, putting question on the effectiveness of Transformer architecture and attention mechanism for time series analysis, especially in the LTSF (Long-term Time Series Forecasting) task. As a fight-back, PatchTST (Nie et al., 2023) improves the capacity of Transformer architecture by introducing patching and channel-independence. Moreover, in CV, Metaformer (Yu et al., 2022a) provides a strong baseline for vision Transformers. It uses a simple pooling operator as the token mixer (which is traditionally implemented by attention mechanism) to aggregates information among tokens and achieves reasonable performance, thereby attributes the model capacity to the Transformer architecture itself.

With all these observations, this paper aims to explore what is really useful for time series transformers. We abstract the essential parts of time series Transformers as MetaTST (**Meta T**ime **S**eries **T**ransformer). MetaTST contain time series tailored components such as decomposition, instance norm as well as patching technique. Meanwhile, it does not specify concrete token mixer. By implementing the token mixer with simple non-parametric operator pooling, we demonstrate that the MetaTST architecture can bring promising performance through extensive experiments on 4 time series analysis tasks.

The contributions of this paper are two-fold. Firstly, this paper summarize the time series transformers into a general architecture MetaTST, and empirically demonstrate that general transformer architecture plus with time series tailored components can achieve promising performance. Secondly, this paper evaluates the proposed MetaTST on different time series tasks including forecasting, classification, imputation and anomaly detection. MetaTST performs on par with other well-acknowledge time series Transformers. Thus, MetaTST can serve as a good start base for future time series Transformer design.

## 2 RELATED WORK

Transformer (Vaswani et al., 2017) is first proposed for NLP tasks and then rapidly become popular in many various tasks such as computer vision (Dosovitskiy et al., 2021) and time series (Li et al., 2019; Zhou et al., 2021). Along the line of transformers for time series analysis, the main challenge of time series Transformer is the quadratic complexity of dot-product attention in self-attention mechanism. In order to tackle this problem, (Zhou et al., 2021) points out that the attention score is sparsely distributed, thereby it is possible to reduce the complexity of attention mechanism while maintaining most information. For example, Autoformer (Wu et al., 2021) propose auto-correlation that can seamlessly replace multi-head attention and be able to capture series-wise dependence of time series. Fedformer (Zhou et al., 2022) capture frequency domain information with Fourier Transform.

The other line of research provides methods on how to incoporate insights of time series into deep learning models especially for Transformers. Multi-level seasonal-trend decomposition is proposed by (Wu et al., 2021) and proved to be a useful design by (Zeng et al., 2023). (Nie et al., 2023) proposes patching to enable the model to directly capture series-wise dependense and keep channal indenpent. (Kim et al., 2022) and (Liu et al., 2022) notice the problem of distribution shift between training and testing dataset. Similar instance normalization is proposed to solve this problem.

However, as questioned by (Zeng et al., 2023), are Transformers effective for time series forecasting? They show that a simple linear model with decomposition can beat many complex Transformer-based models on long-term time series forecasting task. Metaformer (Yu et al., 2022a;b) points out that complex token-mixer (attention) in Transformer can be replaced by a light-weight and simple pooling module while maintaining most of performance. What really matters is the Metaformer architecture that consists of input-embedding, residual connection, arbitrary token-mixer, channel-mixer. This paper, however, aims at verifing is similar hypothesis holds in time series forecasting task: Metaformer plus with add-on time series adopted tricks are all you need for time series forecasting.

## 3 METHOD

### 3.1 THE METATST FRAMEWORK

Figure.1 shows the overall framework of MetaTST. MetaTST is an abstracted general architecture based on transformer with time series related modifications. Note that the token mixer, which is often implemented by various attention mechanisms, is not specified, meaning that any token/time-wise aggregation modules can be applied. Given the input $I$ steps multivariate time series $\mathbf{X} \in \mathbb{R}^{I \times C}$ of $C$ variables, the input is first processed by instance norm module to mitigate the influence of distribution shift between training and testing sets. Then the positional encoding is added and the whole sequence is transformed by patching to make it suitable for Transformers.

After that, the input time series is decomposed into seasonal part and trend part, then fed into the MetaTST encoder stacks. Each stack contains a token mixer to gather time-wise information and a feed forward layer module to gather channal-wise information. Two series deocomposition modules are also included to gradually decompose the time series so that it can be processed better by next module. Note that only the seasonal part goes through these modules, the decomposed trend are aggregated together and added with the seasonal part at the end of the encoder stack. Finally, the extracted features are fed into the projection head, which could be different between generative tasks such as forecasting and identify tasks such as classification. If it is for generative tasks, the output has to be denormalized.

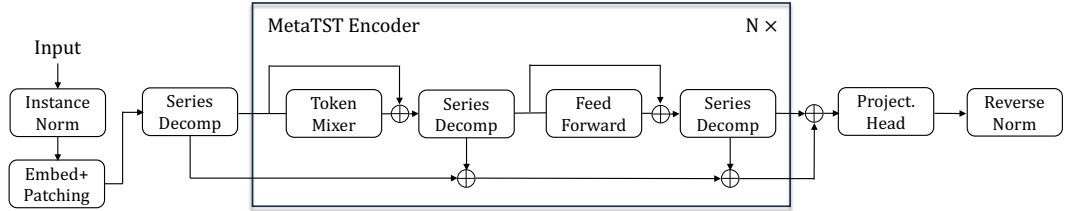

Figure 1: The overall framework of MetaTST.

## 3.2 ESSENTIAL COMPONENTS FOR TIME SERIES TRANSFORMER

**Pooling as Token Mixer.** Token mixer is often implemented by various attention mechanism, such as vanilla attention (Vaswani et al., 2017), autocorrelation (Wu et al., 2021), frequency enhanced block (Zhou et al., 2022) and so on. This line of work often attributes their model capacity to the elaborately designed attention mechanism. In this paper, we use a simple parameter-free operator, i.e. average pooling, to replace the attention. Compared with other attention mechanisms, pooling is extremly simple and the computation cost is rather low. As a token mixer, the receptive field of a single pooling operator cannot cover the whole sequence. Thus, the pooling size is set to be rather large to increase the receptive filed of each pooling layer.

**Decomposition.** Time series often consists of components with different dynamics. For example the house price may grow with years and fluctuate within a year. Thus it is useful to decompose those patterns and process for them respectively. Seasonal-Trend Decomposition Zeng et al. (2023); Wu et al. (2021) has been used in several time series forecasting models. And it is of great importance for their accurate forecasting. Formally, given the input series $\mathbf{X}$, the decomposition module divide it into seasonal part $\mathbf{X}_s$ and trend part $\mathbf{X}_t$. This procedure can be implemented simply via AvgPool1d in PyTorch. Formally,

$$\mathbf{X}_t = \text{AvgPool1d}(\mathbf{X}) \tag{1}$$
$$\mathbf{X}_s = \mathbf{X} - \mathbf{X}_t \tag{2}$$

A time series may contain complicated patterns that cannot be decomposed with only one operation. Thus, it is necessary to do multiple decomposition operation. In MetaTST, global decomposition is conducted firstly to filter out global trend part, so that the encoders only handle the seasonal part.

**Patching.** Patching is first introduced in vision Transformers (Dosovitskiy et al., 2021). It split a input 2D image into local patches so that they can be treated as a sequence by Transformer. Back into time series, this technique is also useful since it can significantly reduces nominal sequence length and eliminate the memory constaints hindering Time Series Transformers to handle long sequences (Nie et al., 2023). Given the orginal input series $\mathbf{X} = \{\mathbf{x}^{(1)}, \mathbf{x}^{(2)}, ..., \mathbf{x}^{(n)}\}$, for each univariate time series $\mathbf{x}^{(i)}$, it is splited into 2D patches with patch length $P$ and stride $S$. Then the patches sequence is $x_P^i \in \mathbb{R}^{P \times N}$ and $N = \frac{L-P}{S} + 2$ is the number of patches. However, with batches and multivariate setting, this prcoess generates a 4D tensor $\mathbf{X}_p \in \mathbb{R}^{B \times C \times P \times N}$. We merge the first two dimension of $\mathbf{X}_p$ and then get $\mathbf{X}_p' \in \mathbb{R}^{(B*C) \times P \times N}$ so that it can be processed by Transformer models.

**Instance Normlaization.** The data distribution between training and test set can be different, leading to degradation of a well trained model performance on test set. The instance norm Kim et al. (2022); Liu et al. (2022) can tackle this problem to some extend. By normalize each input time series instance, and denormalize back the model outputs, it stablizes the value to comply with the distirbution of the test set. Thereby increase the performance on generative tasks such as forecasting, imputation, and anomaly detection (the observation outliers compared with prediction are regarded as anomaly). MetaTST adopts a RevIN layer which makes extra learnable affine transform of the normalized data. Formally, for $k$-th instance, each point $x_{kt}^{(i)}$ in input series at step $t$ is normalized as:

$$\hat{x}_{kt}^{(i)} = \gamma_k \left( \frac{x_{kt}^{(i)} - \mathbb{E}_t\left[x_{kt}^{(i)}\right]}{\sqrt{\text{Var}\left[x_{kt}^{(i)}\right] + \epsilon}} \right) + \beta_k \tag{3}$$

and final prediction is denormalized as:

$$\hat{y}_{kt}^{(i)} = \sqrt{\mathrm{Var}\left[x_{kt}^{(i)}\right] + \epsilon} \cdot \left(\frac{\tilde{y}_{kt}^{(i)} - \beta_k}{\gamma_k}\right) + \mathbb{E}_t\left[x_{kt}^{(i)}\right] \tag{4}$$

where $\gamma_k$ and $\beta_k$ can be fixed or learnable parameters.

## 4 EXPERIMENTS

**Baselines.** Since this paper aims to summarize the effictive components in time series analysis, we compare the performance of MetaTST with several well-acknowledged Transformer-based time series models, including Autoformer (Wu et al., 2021), FEDformer (Zhou et al., 2022), Pyraformer (Liu et al., 2021). Beside, to verify the effectiveness of MetaTST architecture, vanilla Transformer (Vaswani et al., 2017) is taken as baseline as well.

**General Setup.** The model is trained with the ADAM (Kingma & Ba, 2014) optimizer with an initial learning rate of $10^{-3}$. Batch size is set to 32, shrinked if the model runs out of GPU memory under large batch size. The training process is early stopped within 10 epochs for generative tasks including forecasting, imputation and anomaly detection, implemented in PyTorch Paszke et al. (2019) with codebase from (Wu et al., 2022) and conducted on NVIDIA RTX 3090 24GB GPUs. Generally, the time series Transformers have 2 encoder layers and 1 decoder layer. Since the MetaTST does not contain a decoder, for fair comparison, the number of encoder layer in MetaTST is set to 3.

### 4.1 FORECASTING

**Setup.** In order to verify the hypothesis, we conduct empirical experiments of long term forecasting task on ETTm1, Traffic, Weather and ECL datasets Zhou et al. (2021); Wu et al. (2021), as well as short term forecasting task on M4 dataset (Makridakis et al., 2018). Loss function is Mean Squared Error (MSE).

**Results.** Table. and Table. 2. shows the long-term forecasting results and short-term forecasting results respectively. Surprisingly, MetaTST achieve most of the best performance on these benchmarks. For the M4 dataset, MetaTST outperforms all other models, showing that the proposed framework suits the forecasting tasks very well.

Pooling operator aggregates nearly tokens evenly. Thus it is an extremly simple token mixer. However, the experiment results show that with that kind of simple token mixing operator, MetaTST still obtain competative performance compared with other Transformer-based model. Fig. 2 gives show cases of forecasting results on ECL and ETTm1 dataset. Although they are difference quantatively on MSE metric, the actual prediction shows no significant diffrence. This findings conveys that the MetaTST is the base-stone for Transformer models to achieve reasonable performance on time series forecasting task.

### 4.2 IMPUTATION

**Setup.** Missing values often appear in real world time series data due to the malfunction of data collecter. To facilite down stream tasks, it is necessary to recover the original data with the partially missing data. To verify the performance of MetaTST on imputation tast, three typical datasets ETTm1, ECL and Weather are selected. In order to compare the model capacity under different proportions of missing data, the ratio we randomly masked in the experiment varies in $12.5\%, 25\%, 37.5\%, 50\%$.

**Results.** As shown in Table. 3, the MetaTST performs on par with other Transformer-based models. Revealing that the MetaTST architecture is suitable for imputation task.

### 4.3 ANOMALY DETECTION

**Setup.** Detecting anomalies from monitoring data is an important application for various areas. Since anomalies are often hidden in large amounts of data, it is hard to find those anomalies by people. Here we foucus on unsupervised time series anomaly detection. The experiments are conducted

Table 1: Results of the long-term forecasting task

| Dataset | Length | Autoformer | | FEDformer | | Pyraformer | | MetaTST | |
|---|---|---|---|---|---|---|---|---|---|
| | | MSE | MAE | MSE | MAE | MSE | MAE | MSE | MAE |
| ettm1 | 96 | 0.438 | 0.446 | 0.419 | 0.452 | 0.604 | 0.513 | 0.329 | 0.367 |
| | 192 | 0.484 | 0.470 | 0.447 | 0.456 | 0.651 | 0.559 | 0.374 | 0.390 |
| | 336 | 0.464 | 0.475 | 0.443 | 0.456 | 0.779 | 0.653 | 0.402 | 0.409 |
| | 720 | 0.464 | 0.479 | 0.539 | 0.508 | 0.896 | 0.701 | 0.463 | 0.443 |
| traffic | 96 | 0.602 | 0.384 | 0.590 | 0.365 | 0.867 | 0.468 | 0.512 | 0.336 |
| | 192 | 0.605 | 0.371 | 0.600 | 0.369 | 0.869 | 0.467 | 0.509 | 0.332 |
| | 336 | 0.684 | 0.432 | 0.643 | 0.406 | 0.881 | 0.469 | 0.523 | 0.336 |
| | 720 | 0.650 | 0.395 | 0.653 | 0.400 | 0.896 | 0.473 | 0.559 | 0.353 |
| weather | 96 | 0.270 | 0.346 | 0.218 | 0.304 | 0.194 | 0.276 | 0.186 | 0.223 |
| | 192 | 0.305 | 0.369 | 0.275 | 0.347 | 0.227 | 0.312 | 0.230 | 0.260 |
| | 336 | 0.352 | 0.395 | 0.406 | 0.439 | 0.304 | 0.366 | 0.283 | 0.298 |
| | 720 | 0.456 | 0.458 | 0.453 | 0.462 | 0.395 | 0.418 | 0.344 | 0.344 |
| electricity | 96 | 0.234 | 0.342 | 0.193 | 0.310 | 0.386 | 0.449 | 0.170 | 0.259 |
| | 192 | 0.215 | 0.324 | 0.212 | 0.326 | 0.378 | 0.443 | 0.178 | 0.266 |
| | 336 | 0.291 | 0.389 | 0.233 | 0.350 | 0.376 | 0.443 | 0.193 | 0.282 |
| | 720 | 0.296 | 0.391 | 0.268 | 0.377 | 0.376 | 0.445 | 0.233 | 0.315 |

Table 2: Results of the short-term forecasting task in the M4 dataset.

| Period | Metric | Autoformer | FEDformer | Pyraformer | Transformer | PatchTST | MetaTST |
|---|---|---|---|---|---|---|---|
| Year | SMAPE | 69.522 | 17.974 | 13.604 | 14.694 | 13.564 | 13.396 |
| | MASE | 18.142 | 4.062 | 3.075 | 3.304 | 3.050 | 3.005 |
| | OWA | 4.409 | 1.061 | 0.803 | 0.865 | 0.799 | 0.788 |
| Quarterly | SMAPE | 73.760 | 14.485 | 10.610 | 11.506 | 10.791 | 10.805 |
| | MASE | 13.282 | 1.872 | 1.246 | 1.375 | 1.299 | 1.305 |
| | OWA | 8.192 | 1.340 | 0.936 | 1.024 | 0.964 | 0.966 |
| Monthly | SMAPE | 69.837 | 18.235 | 13.887 | 15.589 | 14.540 | 13.262 |
| | MASE | 11.164 | 1.592 | 1.053 | 1.209 | 1.139 | 1.005 |
| | OWA | 7.670 | 1.381 | 0.976 | 1.109 | 1.039 | 0.932 |
| Others | SMAPE | 106.379 | 6.721 | 4.804 | 5.829 | 6.350 | 4.778 |
| | MASE | 82.033 | 4.793 | 3.238 | 4.034 | 4.020 | 3.268 |
| | OWA | 24.129 | 1.463 | 1.016 | 1.249 | 1.302 | 1.018 |
| Average | SMAPE | 72.533 | 16.699 | 12.581 | 13.915 | 13.006 | 12.279 |
| | MASE | 16.821 | 2.388 | 1.674 | 1.872 | 1.761 | 1.650 |
| | OWA | 7.072 | 1.240 | 0.901 | 1.002 | 0.940 | 0.884 |

on five anomaly detection benchmarks including: SMD, MSL, SMAp, SWaT and PSM, covering different applications. Following previous work on this task Xu et al. (2021); Wu et al. (2022), the dataset is splited into consecutive non-overlapping segments by sliding window. And only the classical reconstruction error is regarded as the shared anomaly criterion for all experiments.

**Results.** As shown in Table 4, MetaTST achieves a reasonable performance in anomaly detection task with the mose simple token-mixer. The performance can be attributed to the MetaTST architecture.

## 5 CONCLUSION AND FUTURE WORK

This paper summarizes recent research on time series Transformers by proposing a abstract model architecture called MetaTST. It contains essential components for time series Transformers including the overall architecture, instance normalization, decomposition and patching. Compared with other time series Transformers, MetaTST uses a simple pooling operation but can still achieve competitive results, showing that the capacity of time series Transformers attributes a lot to the whole time-series-adopted architecture. Thus, the hypothesis proposed by Metaformer perhaps holds in time series analysis area. Our work reveals where the capacity of time series Transformers come from. Thus, MetaTST has the potential to be the base model for future model design and serve as

Table 3: Imputation results on Weather, ETTm1 and ECL datasets.

| Dataset | Mask Ratio | Transformer | | Autoformer | | FEDformer | | Pyraformer | | MetaTST | |
|---------|-----------|------|------|------|------|------|------|------|------|------|------|
| | | MSE | MAE | MSE | MAE | MSE | MAE | MSE | MAE | MSE | MAE |
| Weather | 0.125 | 0.033 | 0.087 | 0.357 | 0.438 | 0.044 | 0.107 | 0.030 | 0.074 | 0.031 | 0.057 |
| | 0.250 | 0.035 | 0.086 | 0.144 | 0.252 | 0.055 | 0.128 | 0.036 | 0.089 | 0.033 | 0.057 |
| | 0.375 | 0.039 | 0.097 | 0.135 | 0.239 | 0.076 | 0.159 | 0.039 | 0.091 | 0.034 | 0.058 |
| | 0.500 | 0.042 | 0.094 | 0.180 | 0.281 | 0.116 | 0.211 | 0.041 | 0.092 | 0.038 | 0.063 |
| ETTm1 | 0.125 | 0.023 | 0.107 | 0.718 | 0.699 | 0.034 | 0.130 | 0.032 | 0.128 | 0.046 | 0.143 |
| | 0.250 | 0.028 | 0.117 | 0.526 | 0.573 | 0.053 | 0.163 | 0.035 | 0.132 | 0.055 | 0.150 |
| | 0.375 | 0.035 | 0.130 | 0.350 | 0.443 | 0.083 | 0.202 | 0.041 | 0.140 | 0.060 | 0.159 |
| | 0.500 | 0.044 | 0.145 | 0.313 | 0.402 | 0.133 | 0.260 | 0.048 | 0.152 | 0.067 | 0.167 |
| ECL | 0.125 | 0.150 | 0.278 | 0.191 | 0.328 | 0.185 | 0.323 | 0.190 | 0.303 | 0.059 | 0.163 |
| | 0.250 | 0.157 | 0.282 | 0.198 | 0.309 | 0.207 | 0.340 | 0.216 | 0.346 | 0.072 | 0.183 |
| | 0.375 | 0.168 | 0.290 | 0.216 | 0.346 | 0.225 | 0.355 | 0.195 | 0.305 | 0.088 | 0.203 |
| | 0.500 | 0.180 | 0.297 | 0.234 | 0.360 | 0.251 | 0.372 | 0.207 | 0.312 | 0.108 | 0.227 |

Table 4: Results of anomaly detection task.

| | Transformer | | | Autoformer | | | FEDformer | | | Pyraformer | | | PatchTST | | | MetaTST | | |
|------|------|------|------|------|------|------|------|------|------|------|------|------|------|------|------|------|------|------|
| | P | R | F1 | P | R | F1 | P | R | F1 | P | R | F1 | P | R | F1 | P | R | F1 |
| MSL | 89.98 | 73.79 | 81.09 | 90.53 | 74.96 | 82.01 | 90.71 | 75.41 | 82.35 | 89.01 | 70.84 | 78.90 | 88.31 | 70.77 | 78.57 | 88.51 | 71.64 | 79.19 |
| PSM | 99.36 | 83.20 | 90.56 | 99.99 | 78.96 | 88.24 | 99.98 | 81.94 | 90.07 | 98.53 | 88.36 | 93.17 | 98.84 | 93.54 | 96.12 | 98.73 | 90.91 | 94.66 |
| SMAP | 90.96 | 62.28 | 73.94 | 91.47 | 67.66 | 77.79 | 89.96 | 55.47 | 68.62 | 89.56 | 54.54 | 67.80 | 90.63 | 55.51 | 68.85 | 90.17 | 53.75 | 67.35 |
| SMD | 78.48 | 65.27 | 71.26 | 78.41 | 65.06 | 71.12 | 78.44 | 64.98 | 71.08 | 79.16 | 93.54 | 73.23 | 87.26 | 82.12 | 84.61 | 87.15 | 77.53 | 82.06 |
| SWAT | 99.70 | 66.08 | 79.48 | 99.96 | 65.55 | 79.18 | 99.96 | 65.55 | 79.18 | 99.94 | 65.56 | 79.18 | 91.34 | 83.31 | 87.14 | 91.45 | 84.23 | 87.69 |
| *Avg F1* | | | 79.27 | | | 79.67 | | | 78.26 | | | 78.46 | | | 83.06 | | | 82.19 |

a baseline for new Transformer-based models. Each part of MetaTST is proven to be effective by extensive experiments.

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
