# OpenReview forum: "MetaTST: Essential Transformer Components for Time Series Analysis"
_ICLR.cc/2024/Conference — ICLR 2024 Conference Withdrawn Submission_

### Official Review · Reviewer_ehcv · 2023-10-30

**Soundness:** 1 poor
**Presentation:** 1 poor
**Contribution:** 1 poor
**Rating:** 1
**Confidence:** 5

**Summary:**

The paper implements the Metaformer architecture from the vision domain for time series tasks.

**Strengths:**

None

**Weaknesses:**

1. Paper is poorly written overall, and not up to standard for a conference like ICLR -- the page limit is 9 pages, please consider adding more content such as more detailed formulation of the proposed approach, more empirical analysis and ablations to further understand why the proposed method works.
2. Stronger baselines are omitted in empirical comparison. Importantly, the experiment setup follows the TimesNet paper, but this comparison is missing.

**Questions:**

None

---

### Official Review · Reviewer_U24w · 2023-10-30

**Soundness:** 1 poor
**Presentation:** 2 fair
**Contribution:** 1 poor
**Rating:** 3
**Confidence:** 4

**Summary:**

This article focuses on the self-attention aspect of the Transformer, referencing Metaformer's pooling replacement method for self-attention in the Token Mixer component, and applying it to the proposed temporal task architecture, MetaTST. This approach effectively reduces computation costs and is parameter-free, effectively addressing the computational and memory issues associated with self-attention as the recent works, like Fedformer, Autoformer. Subsequent experiments have also demonstrated that this Token Mixer achieves similar or even better results compared to the baseline method.

**Strengths:**

The authors have attempted some methods from other fields, and the article is clear and easy to understand.

**Weaknesses:**

1. The number of baseline methods is insufficient, and the experimental datasets are limited. The experiments appear to be inadequate, as many necessary ablation experiments and comparisons are missing, such as verifying the effectiveness of using pooling as a replacement for self-attention in large datasets like traffic or electricity.

2. The paper lacks innovation as it seems to be a straightforward transfer of some ideas from MetaFormer into the temporal domain, even though the original work was not in the time series domain. More comprehensive comparisons of various replacement methods for the Token Mixer have been made.

3. Additionally, there is no clear and evident demonstration of the innovation in this multifunctional architecture.

**Questions:**

1、Variety of Token Mixer Methods in Metaformer: Metaformer introduces several methods in the Token Mixer component, including MLP, ConvFormer, IdentityFormer, and the subsequent Inception Mixer. These methods provide a range of options for experimentation.
2、Lack of Persuasiveness in Experiments: The experiments are criticized for lacking persuasiveness in terms of baseline models and dataset diversity. This suggests that the paper may need to include more comprehensive experiments or justify the choices made in the current experimental setup.
3、Performance of Pooling in PatchTST: There is curiosity about how pooling performs when directly applied to newer approaches like PatchTST. This raises questions about the effectiveness and adaptability of pooling within different time series frameworks.

---

### Official Review · Reviewer_5kui · 2023-10-31

**Soundness:** 2 fair
**Presentation:** 1 poor
**Contribution:** 1 poor
**Rating:** 1
**Confidence:** 5

**Summary:**

This paper presents MetaTST, a versatile time series Transformer architecture that combines standard Transformer components with time series-specific features, omitting the traditional token mixer in favor of non-parametric pooling operators. Experients demonstrate the effectiveness.

**Strengths:**

1. this paper studies a classic time series forecasting problem.
2. this paper is easy to understand.

**Weaknesses:**

1. The writing is very worse, which lacks of many technical details.
2. This paper seems to be not finished yet. It should be desk rejected.
3. Experiental results are less convincing.

**Questions:**

See weakness.

---

### Official Review · Reviewer_atVd · 2023-11-01

**Soundness:** 2 fair
**Presentation:** 2 fair
**Contribution:** 2 fair
**Rating:** 3
**Confidence:** 3

**Summary:**

The paper has two primary contributions, including defining the MetaTST architecture and showcasing its empirical success across forecasting, classification, imputation, and anomaly detection tasks. These results establish MetaTST as a robust and adaptable foundation for future time series Transformer designs, raising important questions about the necessity of attention mechanisms in time series analysis.

**Strengths:**

1. This paper is easy to follow.

**Weaknesses:**

1. The main contribution of this article appears to be replacing the self-attention mechanism in the transformer with pooling operations. The contribution is incremental.
2. The experiments are not comprehensive enough, for example, there is a lack of comparison in terms of  computation complexity.

**Questions:**

1. The design motivation of the algorithm in this paper is limited. For example, what considerations led to the assertion that replacing the self-attention mechanism with pooling is beneficial? Why can pooling better handle time series data?
2. Is this algorithm effective for non-stationary and other time series data?
3. More competing methods  should be considered in the experiment.
4. The lower computational cost is an advantage of pooling, so it is recommended to include a comparison of algorithm execution time in the experiments.